# pH Sensitive Pluronic Acid/Agarose-Hydrogels as Controlled Drug Delivery Carriers: Design, Characterization and Toxicity Evaluation

**DOI:** 10.3390/pharmaceutics14061218

**Published:** 2022-06-08

**Authors:** Mariam Aslam, Kashif Barkat, Nadia Shamshad Malik, Mohammed S. Alqahtani, Irfan Anjum, Ikrima Khalid, Ume Ruqia Tulain, Nitasha Gohar, Hajra Zafar, Ana Cláudia Paiva-Santos, Faisal Raza

**Affiliations:** 1Faculty of Pharmacy, The University of Lahore, Lahore 54000, Pakistan; maryyee994@hotmail.com (M.A.); anjuum95@yahoo.com (I.A.); 2Faculty of Pharmacy, Capital University of Science and Technology (CUST), Islamabad 44000, Pakistan; nadiashamshad@gmail.com (N.S.M.); nitasha.gohar@cust.edu.pk (N.G.); 3Department of Pharmaceutics, College of Pharmacy, King Saud University, Riyadh 11451, Saudi Arabia; msaalqahtani@ksu.edu.sa; 4Faculty of Pharmaceutical Sciences, GC University, Faisalabad 38000, Pakistan; ikrima_khalid@yahoo.com; 5Faculty of Pharmacy, University of Sargodha, Sargodha 40100, Pakistan; umeruqia_tulain@yahoo.com; 6School of Pharmacy, Shanghai Jiao Tong University, 800 Dongchuan, Road, Shanghai 200240, China; hajrazafar@sjtu.edu.cn; 7Department of Pharmaceutical Technology, Faculty of Pharmacy, University of Coimbra, 3000-548 Coimbra, Portugal; acsantos@ff.uc.pt; 8REQUIMTE/LAQV, Group of Pharmaceutical Technology, Faculty of Pharmacy, University of Coimbra, 3000-548 Coimbra, Portugal

**Keywords:** agarose, pluronic acid, methacrylic acid, pH sensitive behavior, acute toxicity

## Abstract

The objective of this study was to fabricate and evaluate a pH sensitive cross-linked polymeric network through the free radical polymerization technique for the model drug, cyclophosphamide, used in the treatment of non-Hodgkin’s lymphoma. The Hydrogels were prepared using a polymeric blend of agarose, Pluronic acid, glutaraldehyde, and methacrylic acid. The prepared hydrogels were characterized for drug loading (%), swelling pattern, release behavior, the ingredient’s compatibility, structural evaluation, thermal integrity, and toxicity evaluation in rabbits. The new polymer formation was evident from FTIR findings. The percentage loaded into the hydrogels was in the range of 58.65–75.32%. The developed hydrogels showed significant differences in swelling dynamics and drug release behavior in simulated intestinal fluid (SIF) when compared with simulated gastric fluid (SGF). The drug release was persistent and performed in a controlled manner for up to 24 h. A toxicity study was conducted on white albino rabbits. The developed hydrogels did not show any signs of ocular, skin, or oral toxicity; therefore, these hydrogels can be regarded as safe and potential carriers for controlled drug delivery in biomedical applications.

## 1. Introduction

One of the most convenient routes for drug administration is the oral route, which offers numerous advantages over other routes of drug administration. This route of administration is often chosen because it is simple, it enhances patient compliance and ease over injection, and reduces the costs and potentially improves efficacy [1]. On the other hand, conventional oral drug formulations are associated with several short comings, i.e., fluctuations in plasma drug concentrations, shorter half-lives, frequent drug administration, reduced bioavailability, and undesirable side effects [2]. Hence, controlled drug delivery systems are one of the most promising concepts in drug delivery to overcome the limitations of conventional oral drug formulations. They are designed to deliver drugs at predetermined rates for specified periods of time [3,4].

Cancer is a leading cause of death worldwide, accounting for nearly 10 million deaths in 2020 [5]. At present, cancer treatments include chemotherapy, radiotherapy, surgery, and immunotherapy [6,7]. The extensive application of conventional chemotherapy is restricted because of numerous adverse drug reactions, a poor therapeutic index, and a lack of drug targeting. Currently, most anticancer drugs are administered intravenously. Since chemotherapy regimens are designed to deliver the maximal tolerated dose of cytotoxic drugs, despite the complete and immediate bioavailability of these drugs, the distribution of high concentrations of the drug to normal tissue could be unsafe and toxic [8]. Moreover, intravenous chemotherapy requires a hospital visit, nursing, and palliative care, making this kind of administration inconvenient, painful, and depressive for patients. Hence, oral chemotherapy becomes a very promising alternative to the intravenous therapy [9,10].

For enhancing efficacy, reducing the adverse drug reactions and toxicity of chemotherapeutics, and to improve the life quality of patients, several new oral controlled drug delivery approaches have been proposed in recent years which can attain the selective accumulation of drugs in tumor tissue through active and passive targeting [11,12]. Among them, hydrogels are regarded as promising carriers for various therapeutic moieties including anticancer drugs. Moreover, in comparison to normal tissue, cancer tissues have many unique features (e.g., hypoxia, altered pH, and variable permeability and retaining effects), which can be used to design drug delivery vehicles in the form of hydrogels. Hence, traditional chemotherapy drugs are experiencing a resurgence with the advances in the hydrogel-based drug delivery system [13,14,15].

Hydrogel is a three-dimensional polymeric network that can imbibe large amounts of water or biological fluids while maintaining physical integrity, and it is made insoluble due to the presence of chemical and/or physical cross-links [16]. Various properties of hydrogel delivery systems, i.e., tunable release properties, stimuli-responsive behavior, the protection of labile drugs from degradation, and their biocompatibility and biodegradability make them capable of providing therapeutically beneficial results in drug delivery [17].

Stimuli-responsive hydrogels are of great interest since they display reversible swelling behavior in response to external stimuli such as temperature, light, electric fields, humidity, and pH. Among stimuli-responsive hydrogels, pH-sensitive hydrogels are composed of polymers holding pendant acidic or basic groups that either accept or donate a proton in response to the change in pH of the external environment. Hence, pH-sensitive hydrogels can display alterations in their characteristics because of the variations in pH that are known to occur at several body sites such as the gastrointestinal tract [18,19,20].

Generally, natural polymers that form hydrogels are obtained from protein polymers, such as elastin, collagen, or polysaccharide polymers such as glycosaminoglycans, alginate, or agarose [21]. Agarose is a natural oxygen rich polysaccharide. It is obtained from seaweed. It has 1,3-linked β-D-galactose and 1,4-linked 3,6-anhydro-α-L-galactose [22]. Agarose is popular for its gelation property. After cooling the hot aqueous solution of agarose, a gel is formed which is stable at pH range from 3–9. Agarose is a very effective polymer for making of cross-linking networks with different polymers [23].

Pluronic acid (PF127) is a copolymer of synthetic origin. PF127 fits in the family of PEO-PPO-PEO tri-block copolymers. Its hydrophobic nature is because of its central Polypropylene oxide whereas on both sides the presence of polyethylene oxide makes it hydrophilic. The nature of Pluronic acid is non-toxic. Water dissipation from micellar cores of Pluronic F127 at body temperature contributes majorly in its ability to make gel [24]. When the ambient temperature is below or normal Pluronic F127 solution remains free-flowing. However, alone, PF127 has insufficient mechanical strength and stability, making it unsuitable to be used alone for certain pharmaceutical applications. Therefore, modification and cross linking is required to increase the mechanical strength and attain desired properties of hydrogels [25,26,27].

Methacrylic acid is extensively used in the fabrication of pH-sensitive hydrogels. It is present in the form of a colorless liquid or crystals. It has a hydrophilic nature and an unpleasant odor. It is pH-sensitive and demonstrates extreme variations in swelling behavior when there is a change in pH and ionic strength. Its pH-sensitive behavior grants it a very significant part in temporal or spatial delivery, sustained release delivery systems, evolving controlled release, and using diverse categories of drug moieties [28].

Cyclophosphamide (CP) is an alkylating anti-cancer drug, mainly used for non-Hodgkin’s lymphoma treatment. It can work efficiently due to its high selectivity and toxicity to effected cells [29]. It may be given as an IV bolus or given per oral. CP is an oxazaphosphorine pro-drug [14]. It metabolizes in the liver via the CYP450 enzyme. CP is converted to 4-hydroxycyclophosphamide (4-OHCP) before producing anti-cancer effects [30,31].

Hence, in the current research, oral controlled release pH-sensitive hydrogels have been developed with the primary goal of optimizing drug efficacy, lessening adverse drug reactions, reducing dosage frequency, and enhancing the compliance of patients. The blending of natural and synthetic polymers was conducted to develop hydrogels using cyclophosphamide as the model drug. The pH-dependent monomer was used to impart pH-dependent swelling and drug release behavior. Hydrogels were analyzed for swelling studies and drug release behavior. DSC, TGA, FTIR, PXRD, SEM, and an acute oral toxicity study was also performed.

## 2. Materials and Method

### 2.1. Chemicals

Agarose (AGA) was purchased from Bio Plus Fine Research Chemicals. Pluronic Acid (F127), Benzoyl Peroxide (BPO), and Ethanol were purchased from Sigma Aldrich. Eschenstr., Taufkirchen, Germany, Methacrylic Acid (MAA) was purchased from DAE JUNG Chemical and Metals Co. Ltd, Gyeonggi-do, Korea.; Glutaraldehyde (GA) was purchased from Merck, Frankfurter Strasse, Darmstadt, Germany. The Cyclophosphamide drug was obtained from Suzho Unit Phrmaceutical, Suzho, China. Distilled water was prepared freshly in Research Lab, The University of Lahore, Lahore, Pakistan.

#### Pluronic Acid/Agarose-co-Poly (Methacrylic Acid) Hydrogels

Free radical polymerization technique was used for fabrication of hydrogels by utilizing benzoyl peroxide (BPO) as initiator. Glutaraldehyde was used as cross-linker. Desired quantities of all excipients (monomer, polymers. Initiator, and cross-linker) as mentioned in Table 1 were correctly measured on electric weighing balance. Predefined quantities of distilled water were used to dissolve agarose and Pluronic acid separately using hot plate magnetic stirrer. Then, both solutions were mixed. After this, weighed quantities of monomer (MAA) and initiator (BPO) were mixed and then combined with the previous polymeric solution. Mixture was stirred thoroughly on magnetic stirrer. At the end, glutaraldehyde was supplemented dropwise to the whole blend and stirred for 4 h. To remove trapped oxygen, nitrogen stream was purged. Mixture was shifted to the glass tubes and then placed in electric water bath for about 24 h at different temperatures (55 °C for 4 h, 60 °C for 4 h, and 65 °C for 16 h). Upon solidification, test tubes containing hydrogels were taken out of electric water bath and subjected to cooling at room temperature. Developed hydrogels were removed from test tubes, sliced, and thoroughly washed using 50% ethanolic solution to remove any traces of unbounded polymers or monomers. Then, sliced hydrogels were dried by placing them at 40 °C for 24 h in oven. Formulations of varied concentrations of monomers, polymers, and cross-linkers were fabricated to regulate the effect of variables of formulation on drug release [32,33,34]. Proposed structure for developed hydrogel is elaborated in Figure 1.

### 2.2. Drug Loading

Drug was loaded in developed hydrogels through swelling diffusion method. Prepared and dried hydrogel discs of identified weight were immersed in 1 % drug solution in 0.2 M phosphate buffer of pH 7.4 at room temperature for 48 h. After 48 h, swollen disc was taken out of 1% drug solution, washed, and freeze dried in lyophilizer at −55 °C [35,36,37,38].

### 2.3. Characterization

#### 2.3.1. Physical Appearance of Prepared Hydrogels

Hydrogel discs were examined for their physical appearance, particularly their “transparency”. They were also examined for their texture (soft or hard).

#### 2.3.2. Hydrogels Swelling Study at Different pH

Hydrogel swelling study at different pHs was carried out to find pH-responsive behavior of fabricated hydrogels. Firstly, initial weights of hydrogel discs in dried form were noted and then simulated gastric fluid (pH 1.2) and simulated intestinal fluid (pH 7.4) were used to place hydrogel discs at 37 °C [39]. Solution volume was kept at 100 mL. Weight of discs was recorded regularly at defined time intervals until constant weights were achieved [40,41,42].

Equation (1) given below was utilized to calculate the swelling ratio represented by “S” [43,44].
(1)S=(Ws – Wd) / Wd ×100
where Ws shows swollen hydrogels’ mass on programmed time interval and Wd shows the dried hydrogels weight.

#### 2.3.3. Drug Loading Efficiency (DLE), Release Behavior, and Release Kinetics

In a mortar and pestle, drug loaded hydrogel discs were crushed cautiously to gauge drug loading efficiency. The crushed hydrogel discs were weighed and kept in phosphate buffer solution at pH 7.4 at 37 °C. Constant Stirring was performed for 24 h. Drug solution was centrifuged out at 3000 rpm to isolate the supernatant layer, filtered, and then analyzed for drug by means of HPLC-UV, with a C18 column (5 μm, 150 × 4 mm) and detection at wavelength of 195 nm. *DLE* (%) was calculated using Equation (2).
(2)$$DLE=\frac{Actual\;loading}{Theoratical\;loading}\times 100$$

In vitro determination of cyclophosphamide release was carried out to evaluate the drug release pattern as a function of pH. USP dissolution apparatus II was selected to perform dissolution. Simulated gastric and simulated intestinal fluid of pH 1.2 and pH 7.4, respectively, were used as dissolution mediums. Drug loaded hydrogel discs were first weighed and then placed in both mediums. Medium volume was kept at 900 mL in each basket. Temperature was set at 37 °C. Paddle revolution speed (50 rpm) was adjusted. At defined intervals, samples from bucket were drawn, diluted with freshly prepared buffer solution, and then studied by HPLC-UV spectrophotometer at 195 nm wavelength [45]. Release kinetics of drug from formulated hydrogel were examined by figuring numerous kinetic models such as zero-order, first-order, Higuchi, and Korsmeyer–Peppas models [46,47,48]. Mathematical models are helpful to illustrate and additionally enlighten release mechanisms. The foremost suitable model equations used for release kinetics for hydrogels are:

The equation for zero-order kinetics (3):M_t_ M_∞_ = kt(3)

The equation for first-order kinetics (4):ln M_t_ M_∞_ = kt(4)

The equation of Peppas or Korsmeyer–Peppas kinetics (5):M_t_ M_∞_ = kt_n_(5)

The Higuchi Equation (6):M_t_ M_∞_ = kt_1/2_(6)
where M_t_ is the overall quantity of drug released at a time t, M_∞_ is the overall quantity of drug cargo to be freed, n is the diffusivity exponent, and k is the kinetics constant (5).

#### 2.3.4. FTIR Analysis

Fourier Transform Infrared Spectroscopic study was carried out on active ingredient as well as formulation excipients (polymers, monomer, and cross-linker) using Alpha Bruker-Germany, ECO-ATR [49]. Samples were crushed and loaded on crystal spot with the help of spatula and pressure was applied to form compact disc for analysis and then scanned between defined ranges of 4000–400 cm^−1^ [50,51].

#### 2.3.5. Thermogravimetric Analysis (TGA) and Differential Scanning Calorimetry (DSC) Analysis

TGA study of agarose, Pluronic acid, and formulation was carried out to determine physicochemical properties and transition temperature of all samples. Sample of 3–5 mg was placed in a hermetically sealed condition in an aluminum pan. Temperature range of about 0–500 °C was adjusted to heat the samples at 20 °C/min beneath dry nitrogen. Adjusted flow rate was 10 mL/min [52].

DSC study of agarose, Pluronic acid, and formulation was performed to determine the thermal performance of materials related to their association states, structures, and hydrophilic properties. Thermograms of samples were documented using a DSC (8500, Perkin Elmer, American Fork, UT, USA). Nitrogenous atmosphere (20 mL/min) was used to carry out the measurements at heating rate of 10 °C/min and temperature range of about 30–600 °C [53].

#### 2.3.6. Powder X-ray Diffraction (PXRD) Analysis

PXRD was performed on pure drug as well as on formulation by means of XR-diffractometer (Siemens D-500). Material was finely ground and homogenized to determine the crystallographic density and phase identification of a crystal structure. Applied voltage was 40 KV, current of 28 mA, and scanning range used was 2Ɵ with interval of 0°–70° [54,55,56,57,58].

#### 2.3.7. Scanning Electron Microscopy (SEM) Analysis

SEM analysis was conducted on formulation for determination of surface morphology using scanning electron microscope (SEM) (JSM-5910, Pleasonto, CA, USA). Dried crushed sample was placed on aluminum pan coated with double adhesive tape. Gold coater was used to coat sample at 20 mA for 2 min. Scanning was completed in 600 cm^−1^ to 4000 cm^−1^ wavelength range [59,60].

#### 2.3.8. Acute Oral Toxicity Study

The experimental procedure used in the current research was revised and approved by Research Ethics Committee of The University of Lahore, Pakistan (13–2020/REC). Acute toxicity study of Pluronic acid/agarose-co-poly (methacrylic acid) hydrogel was conducted according to organization of economic cooperation and development (OECD) guidelines.

Based on maximum drug entrapment efficiency and in vitro cumulative drug release, one hydrogel formulation was chosen for safety evaluation via acute oral toxicity study. Six adult healthy albino male rabbits (age 3 months, weighing 2.5–3.00 kg) were obtained and divided into two groups. One was labelled “Controlled” and other as “Test group”. The animals were acclimatized on standard laboratory diet for 7 days before testing. OECD approved method used for acute toxicity study was maximum tolerance dose method. Group A was considered as control group and was administered CP in suspension form by adding CP powder to normal saline. Group B was considered as experimental group and administered hydrogel by crushing in mortar and pestle, creating hydrogel dispersion in deionized water, and administrating through oral gavage at 2 g/kg body weight dose.

Both groups were subjected to fasting overnight before every dose. Physical changes were observed for fourteen days. At 15th day, blood samples were drawn for biochemical and hematological analysis. For histopathological analysis, rabbits’ necropsies were completed, and vital organs were removed, cleaned, weighed, and preserved in 10% formaldehyde solution. Guidelines established in guide for Care and Use of Laboratory Animals and animal welfare act were strictly followed.

#### 2.3.9. Statistical Analysis

For statistical analysis, IBM SPSS Statistics 20 program was used. Student’s t-test was used to determine the statistically significant differences between the swelling dynamics and drug release behavior. For acute toxicity study, the variations between two groups were determined by one-way analysis of variance (ANOVA) with Tukey test. A value of *p* < 0.05 was regarded statistically significant.

## 3. Result and Discussions

### 3.1. Physical Appearance of Prepared Hydrogels

A chemically cross-linked Pluronic acid/agarose-co-poly (methacrylic acid) hydrogel blend was prepared through an aqueous free radical polymerization. The texture of the prepared hydrogel discs was rubbery. The formulated hydrogels were white in color as shown in Figure 2. The hydrogel discs exhibited a high mechanical strength before and after swelling and maintained their shape.

#### 3.1.1. Swelling Dynamics

As shown in Figure 3, the developed hydrogels indicated pH-dependent swelling behavior. The pH-dependent behavior in the hydrogels was due to the existence of the MAA monomer. Hence, in SGF at pH 1.2, the presence of a higher number of protonated COOH groups from MAA lowered the ionic repulsion, supported the hydrogen bonding among the COOH groups, and caused the hydrogels to shrink [61,62]. However, the swelling of the prepared hydrogels increased markedly in SIF as compared to SGF. This behavior might be due to the fact that in SIF, the electrostatic repulsion between the carboxylate anions (COO^−^) of MAA and the osmotic swelling force within the hydrogel network causes expansion and the swelling of the hydrogels [63]. Thus, the ionic repulsion between the unprotonated carboxyl groups in the developed hydrogels presented increased swelling kinetics at an alkaline pH, demonstrating that the interrelated spongy network of pH-sensitive Pluronic acid/agarose-co-poly (methacrylic acid) hydrogels gathered additional water molecules in their structure in the swollen state. Additionally, these carboxylate anions (COO^−^) in the polymeric network are inclined towards sturdier solvation, leading to the increased swelling of the hydrogels at pH 7.4 when compared to pH 1.2 [64,65,66]. The statistical evaluation of the differences between the swelling dynamics at pH 1.2 and pH 7.4 was performed by the Student’s *t*-test. The results suggest that there is a significant difference in drug release at pH 1.2 and pH 7.4 (*p* < 0.05). At pH 1.2, the drug release percentage was minimum, whereas at pH 7.4, the drug release percentage was at its maximum.

The effect of the different concentrations of reactants on the swelling behavior of the synthesized hydrogels have been evaluated at pH 1.2 and 7.4 at temperature 37 °C as shown in Figure 4.

When increasing the concentration of agarose from 0.4% to 0.8% and 1.2%, the swelling of the hydrogel increased. The hydrogel with the ratio of 1.2% agarose showed the highest swelling ratio. This might be attributed to the hydrophilic nature of agarose leading to maximum swelling [59,60].

The swelling of the hydrogels reduced when increasing the concentration of Pluronic acid from 0.4% to 0.8% and 1.2%. The Hydrogel with a ratio of Pluronic acid of 0.4% showed the highest swelling. This might be due to fact that upon an increase in the Pluronic acid concentration in the polymeric network, there is also an increase in the viscosity of the reaction system leading to the constrained activities of macroradicals, thus limiting the chain transfer reaction, leading to a noticeable decline in the swelling ratio [67].

The outcome of monomer concentration on the swelling size of the hydrogel was inspected by using variable ratios of MAA in the range of 20%, 24%, and 28%. The formulation with the lowest concentration of MAA showed the maximum swelling. A possible explanation of this behavior might be that increasing the monomer concentration caused an increase in the cross-linking mass of the polymeric network and lead to the reduced flexibility of the polymeric chains. Hence, the swelling ratio was decreased [68].

Upon increasing the concentration of Glutaraldehyde from 0.2% to 0.4% and 0.6%, the swelling of the hydrogel decreased. The formulation with the lowest concentration of glutaraldehyde showed the maximum swelling [69]. The high cross link density at a higher concentration of cross linkers delays the polymeric network’s ability to make hydrogen bonds with water, hence restricting its water absorbing ability and eventually the swelling index of the hydrogel.

#### 3.1.2. Cyclophosphamide Release from Hydrogels and Kinetic Models

Table 2 shows the drug loading and percent drug release from the hydrogel formulation. It was detected that % DLE of the hydrogels verified the hydrogels’ dependence on the swelling behavior and the extension of crosslinking. Hydrogel AGA3 exhibited maximum DLE, whereas AGA7 exhibited minimum DLE. Moreover, the hydrogels’ mean cumulative drug release at pH 1.2 and 7.4 is shown in Figure 5. The statistical evaluation of the differences between drug release at pH 1.2 and pH 7.4 was performed by a Student’s *t*-test. The cyclophosphamide release from the hydrogels was found to be pH-dependent, thus signifying its pH-sensitive behavior. The results suggest that there is a significant difference in drug release at pH 1.2 and pH 7.4 (*p* < 0.05). At pH 1.2, drug release was at its minimum whereas at pH 7.4, the percentage drug release was at its maximum.

Varied proportions of polymers, monomers, and cross-linkers were used to determine the effect of the concentrations on the release behavior of the drug. The drug release behavior was consistent with the swelling dynamics. The Maximum amount of the drug was released at the highest ratio of agarose. This is because a higher concentration of agarose increases the swelling ability and thus increases the penetration and release of the drug through diffusion. However, CP release decreased after increasing the ratio of Pluronic acid, similar to the swelling behavior. A higher concentration of monomers and cross-linkers showed the opposite effect as a greater concentration produces strong bonding and dense polymeric network. The Denser polymeric network resulted in poor swelling and decreased drug release.

After applying kinetic models through DD solver software, the R^2^ (regression coefficient) of the varied fabricated hydrogels was found to support the Korsmeyer–Pappas Model, Higuchi Model, and Zero Order Models. The Zero, Higuchi, and Korsmeyer–Peppas models were nominated based on the principles defined by the goodness of fit which implies the correlation coefficient (R^2^) is close to 1 [70,71]. The manifold release mechanism responsible for active release is the swelling of the polymer matrix, the chemical interaction of drug–polymer, erosion, and diffusion [72]. All formulations agreed with the Fick’s law. The R^2^ values of all formulations are presented in Table 3.

In this study, we have attempted to develop hydrogels via the free radical polymerization technique. The hydrogel formulation AGA3 is considered the most optimized formulation based on the results of the swelling dynamics, drug loading efficiency, and drug release behavior, and has been selected for carrying out the characterization studies as discussed below.

#### 3.1.3. FTIR Analysis

To confirm the grafting and chemical structure of the recently developed hydrogels and the distinct components, FTIR spectroscopy was used as shown in Figure 6. In the FTIR of Pluronic acid (a), the bending of the (O–H) group was observed at approximately 1400 cm^−1^ spectrum of Pluronic acid 2900 cm⁻^1^ (C–H stretch), 1278 cm⁻^1^ (C–O–C stretches), and 1097 cm⁻^1^ (C–O stretch). Similarly, FTIR of agarose (b) showed stretching peaks at approximately 1050 cm^−1^ representing (C-O), 1600 cm⁻^1^ representing (O–H), 3400 cm⁻^1^ representing (N–H), and (C=C), 470 cm⁻^1^ representing (C–H) group. The methacrylic acid spectrum (c) showed stretching peaks at approximately 2930 cm⁻^1^ representing (N–H) and 1296 cm⁻^1^ representing (S=O) group. Bending peaks were observed at approximately 1689 cm⁻^1^, 1632 cm⁻^1^, and at 1425 cm⁻^1^ representing (C–H) group. The FTIR spectrum of cyclophosphamide (d) exhibited stretching at 3000 cm⁻^1^, 1100 cm⁻^1^, and 840 cm⁻^1^ on behalf of (N–H), (C–O), and (C–Cl), correspondingly. The unloaded formulation spectrum (e) depicted peaks at 1350–1480 cm⁻^1^ and 1000–1150 cm⁻^1^ due to the C–H bending and C–O stretching vibration. The intensity of the C–H bending and the C–O stretching vibrations can be explained by the side reduction and to some extent shifting to the lower wavenumber in the hydrogel compared to the original agarose, Pluronic acid, and methacrylic acid. This can be linked with the H-bonding between the hydroxyl (–OH) groups of Pluronic acid, agarose, and methacrylic acid, confirmed grafting. Moreover, in drug loaded hydrogel (f), the spectra showed the presence of all peaks of the pure components with a slight shifting or overlapping indicating the persistently intact drug inside the network with a good chemical stability and compatibility [11,73,74].

#### 3.1.4. Differential Scanning Calorimetry (DSC) Analysis

Figure 7 indicates the DSC Analysis of (a) Agarose, (b) Pluronic Acid, and (c) Optimized hydrogel formulation. The DSC thermogram of agarose (a) displayed a preliminary endothermic peak at 78.41 °C which showed a loss of moisture and a removal of additional volatile components. Two exothermic peaks at about 95.92 °C and 443.68 °C were detected. The first peak could be credited to the glass transition temperature Tg whereas the second peak implies the oxidative degradation of the polymer. The thermogram of Pluronic acid (b) showed endothermic peaks at approximately 65 °C and 170 °C and exothermic peaks at about 98 °C and at 485 °C, respectively. The thermogram of the optimized hydrogel (c) showed an initial endothermic peak at 76.36 and later one at 334.59 °C. Moreover, two exothermic peaks were observed at approximately 258.66 °C and at 471.38 °C. The shifting of the exothermic peaks in developed optimized formulations at higher temperatures when compared to individual components is an indication that the formulation is more stable compared to individual components [75].

#### 3.1.5. Thermogravimetric Analysis (TGA)

Figure 8 indicates the TGA pattern of (a) Agarose, (b) Pluronic Acid, and (c) the Optimized hydrogel formulation. The TGA of agarose showed an initial weight loss of 17.48% at 148.54 °C and a 25% weight loss at 375 °C, respectively. For Pluronic acid, an initial weight loss of 63.31% was observed between 258.81–400.94 °C and a 75% weight loss was observed between 425–480 °C, respectively. However, hydrogel formulation was stable up to 425.12 °C, showing an increase in the degradation half-life of hydrogel when compared to the degradation half-lives of the individual components. Hence, it is evident that the formulation is thermally more stable than pure reactants [76].

#### 3.1.6. Scanning Electron Microscopy (SEM) Analysis

The microphotographs taken by the SEM (Figure 9) presented a wavy and highly rough surface with characteristic creases and cracks. The rough surface may be produced by the fractional collapsing of the polymeric gel network throughout the process of drying. Moreover, hydrogels revealed a high crosslinked density of polymeric networks, having pores and channels for drug entrapment. The porous structure stimulates the surface phenomena of accepting media for eventual swelling and drug release from the formulation [77,78]. The pore structure was retained due to the freezing of the network structure during the lyophilization process which avoids pore shrinking.

#### 3.1.7. X-ray Diffraction (XRD) Analysis

Figure 10 indicates the XRD analysis of (a) Agarose, (b) Pluronic Acid, and (c) the Optimized Hydrogel formulation. The graph of agarose and Pluronic acid showed sharp peaks at 12.75° and 18.3° as well as 18° and 23°, respectively, which showed the crystalline behavior of the polymer. The Optimized Hydrogel formulation also showed sharp peak at 13.5°, representing the crystalline nature of the drug in hydrogel discs. Hence, we can say that the drug remains intact in crystalline form and stable in a developed polymeric network [79].

#### 3.1.8. Acute Oral Toxicity Study

The clinical findings of the control and test group rabbits are shown in Table 4. The clinical findings did not expose any noteworthy fluctuations in the food consumption, water, or body weights of the rabbits [80]. No morbidity or mortality was reported in any group throughout the acute oral toxicity study. The results of the hematology and biochemical analysis of the control and test group rabbits are shown in Table 5. The complete analysis of the blood indicates that all the values were found within a normal range [81]. Moreover, as shown in Figure 11, histopathological studies revealed no significant alterations or abnormality in the tissue structures of the rabbits treated with hydrogel when compared with the control groups [82]. Hence, hydrogels could be considered non-toxic and biocompatible as no specific changes were seen between the test and control group.

## 4. Conclusions

A chemically cross-linked Pluronic acid/Agarose-co-poly (Methacrylic acid) hydrogel blend was successfully formulated and loaded with cyclophosphamide. The formulations were more thermally stable than the individual ingredients. The prepared hydrogels displayed pH-responsive swelling and drug release behavior. The maximum drug release was observed in SIF. The fabricated hydrogels followed the Higuchi, Korsmeyer–Peppas, and zero order release models. Blood sample findings and photomicrographs of the vital organs showed that the hydrogels were biocompatible and non-toxic. Hence, the anticipated method could be used to develop hydrogels with substantial pH-sensitive behavior. The authors’ future aim is to evaluate the in vitro and in vivo cytotoxic activity of the developed formulations. This research will provide close predictions of in vitro and in vivo efficacy, can bridge the gap between in vitro and in vivo anticancer drug evaluations, and provides a better understanding of the design and development of drug delivery systems.

## Figures and Tables

**Figure 1 pharmaceutics-14-01218-f001:**
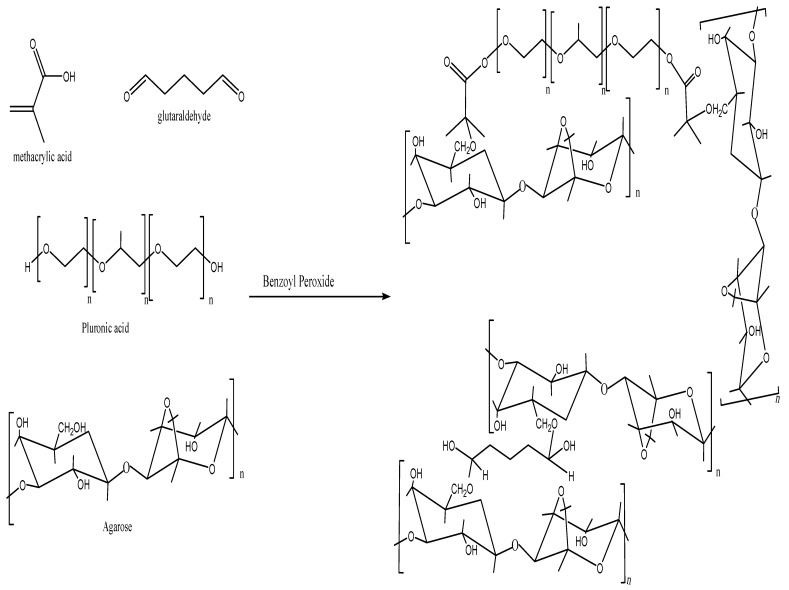
Proposed structure for developed hydrogel.

**Figure 2 pharmaceutics-14-01218-f002:**
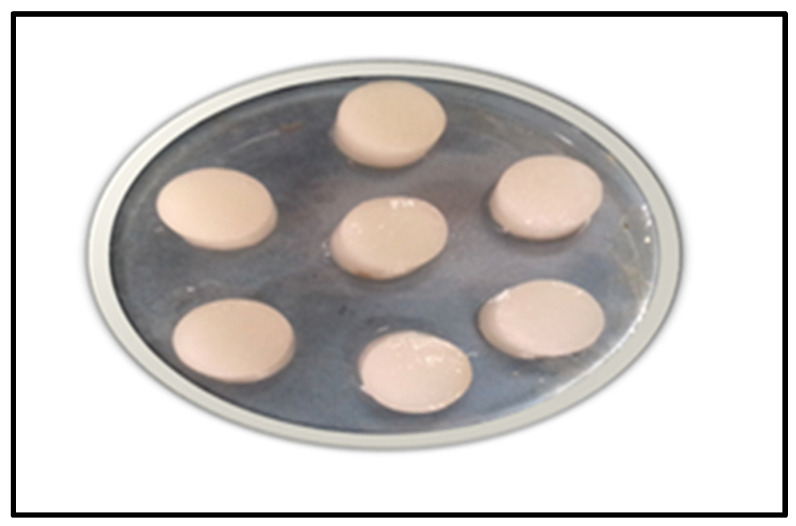
Pluronic acid/agarose-co-poly (methacrylic acid) hydrogels.

**Figure 3 pharmaceutics-14-01218-f003:**
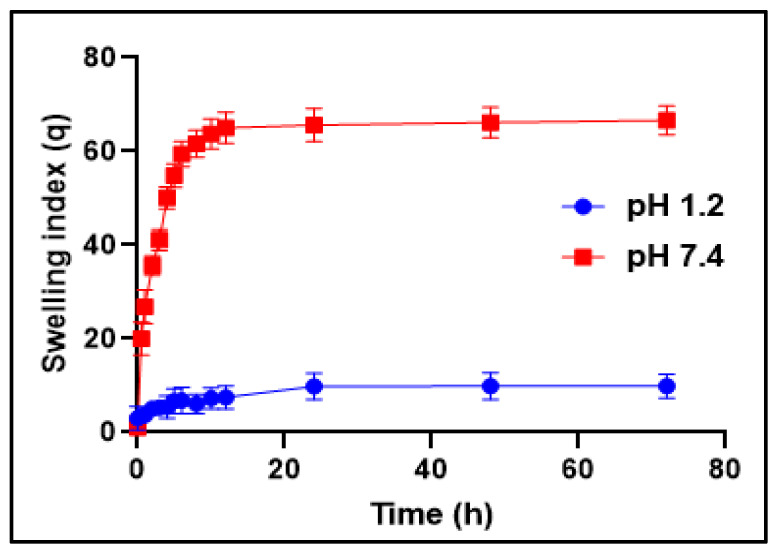
Mean swelling index of hydrogels at pH 1.2 and 7.4.

**Figure 4 pharmaceutics-14-01218-f004:**
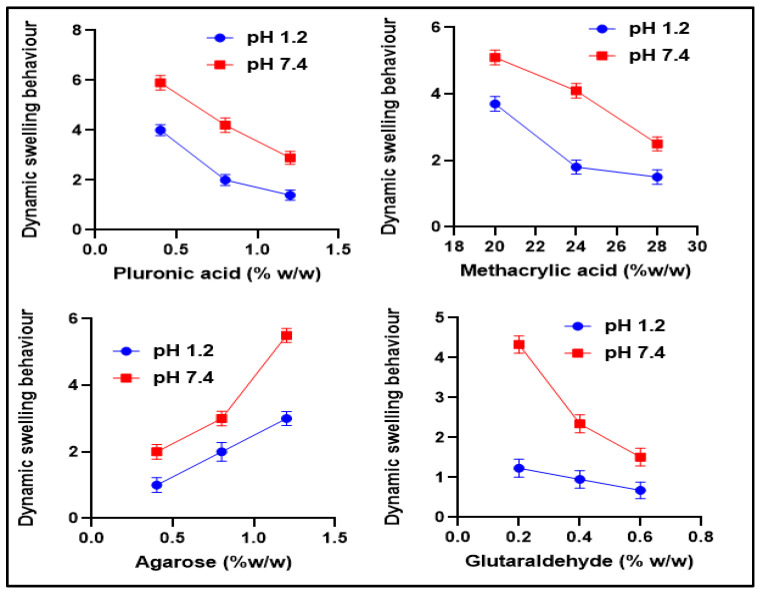
Dynamic swelling coefficient of hydrogels with different concentration of reactants at pH 1.2 and pH 7.4.

**Figure 5 pharmaceutics-14-01218-f005:**
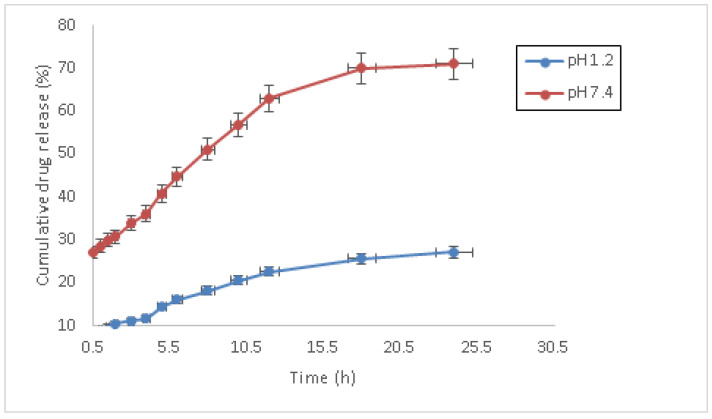
Mean cumulative drug release from hydrogels at pH 1.2 and 7.4.

**Figure 6 pharmaceutics-14-01218-f006:**
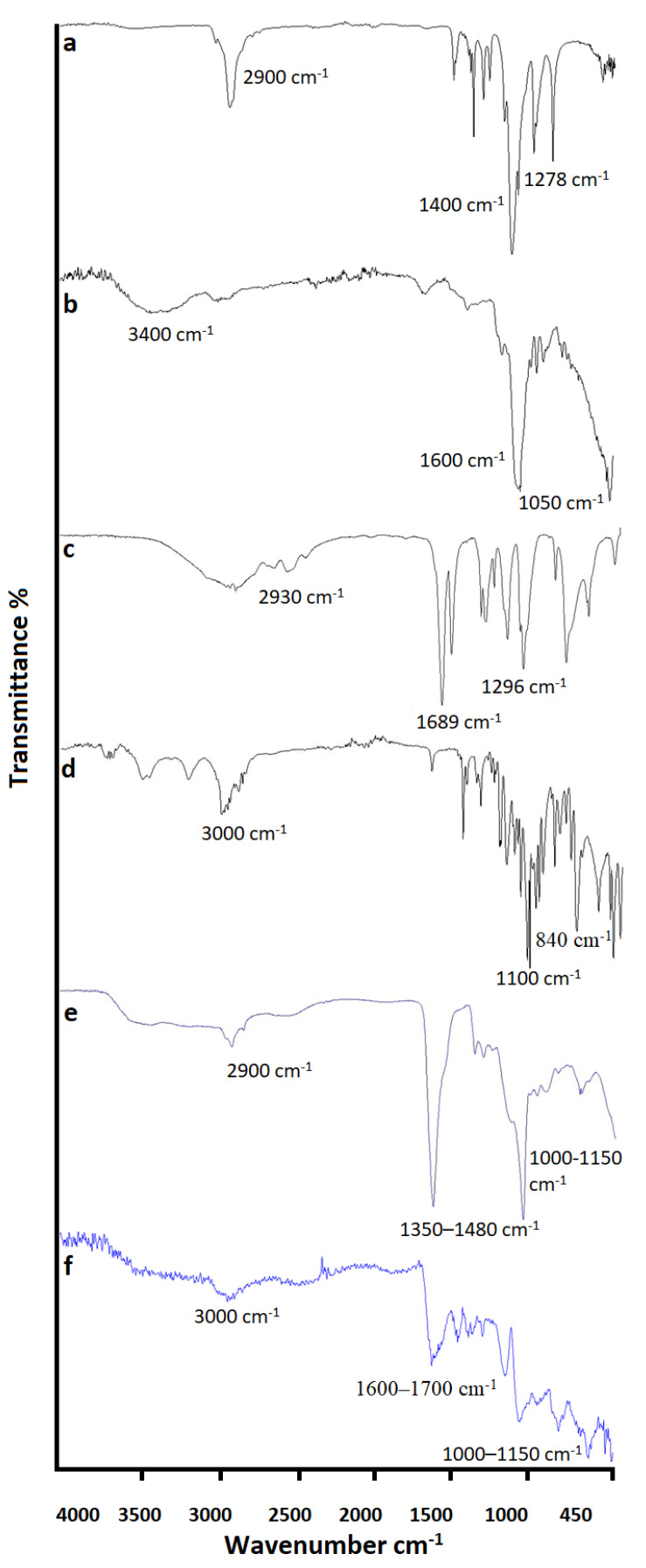
FTIR Analysis of Pluronic acid (**a**), Agarose (**b**), MAA (**c**), Drug (**d**), Unloaded hydrogel (**e**), and Drug loaded hydrogel formulation (**f**).

**Figure 7 pharmaceutics-14-01218-f007:**
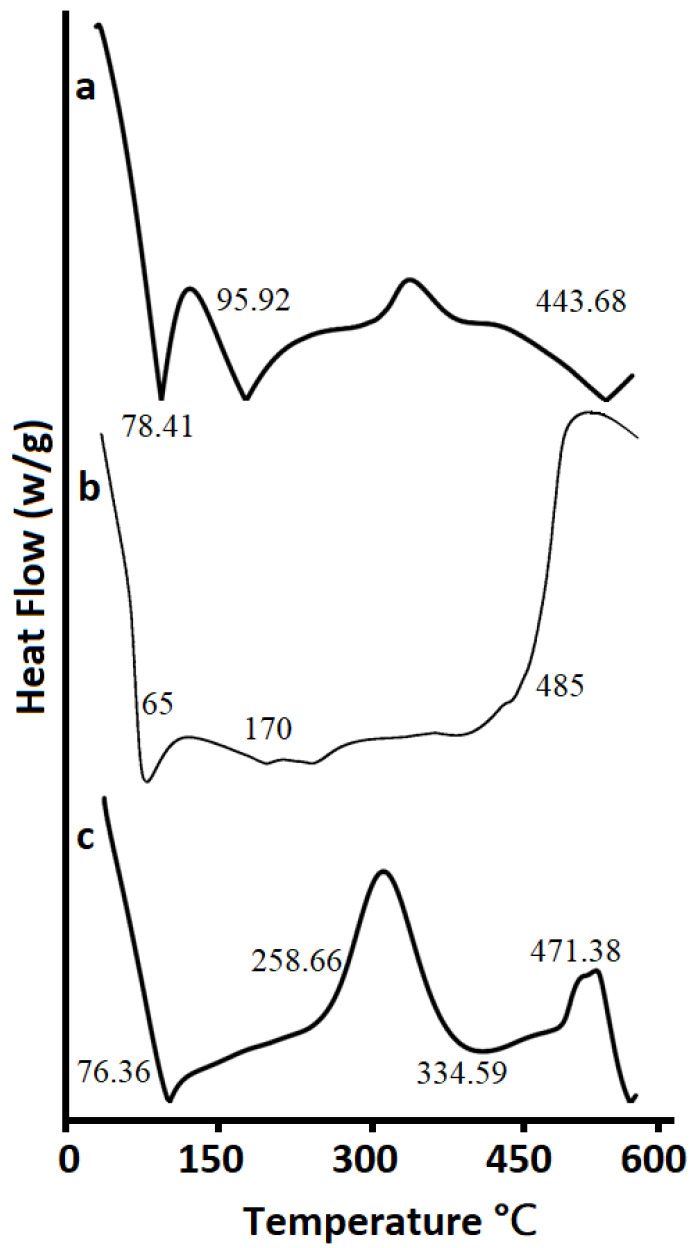
DSC Analysis of (**a**) Agarose, (**b**) Pluronic Acid, and (**c**) Optimized unloaded Formulation.

**Figure 8 pharmaceutics-14-01218-f008:**
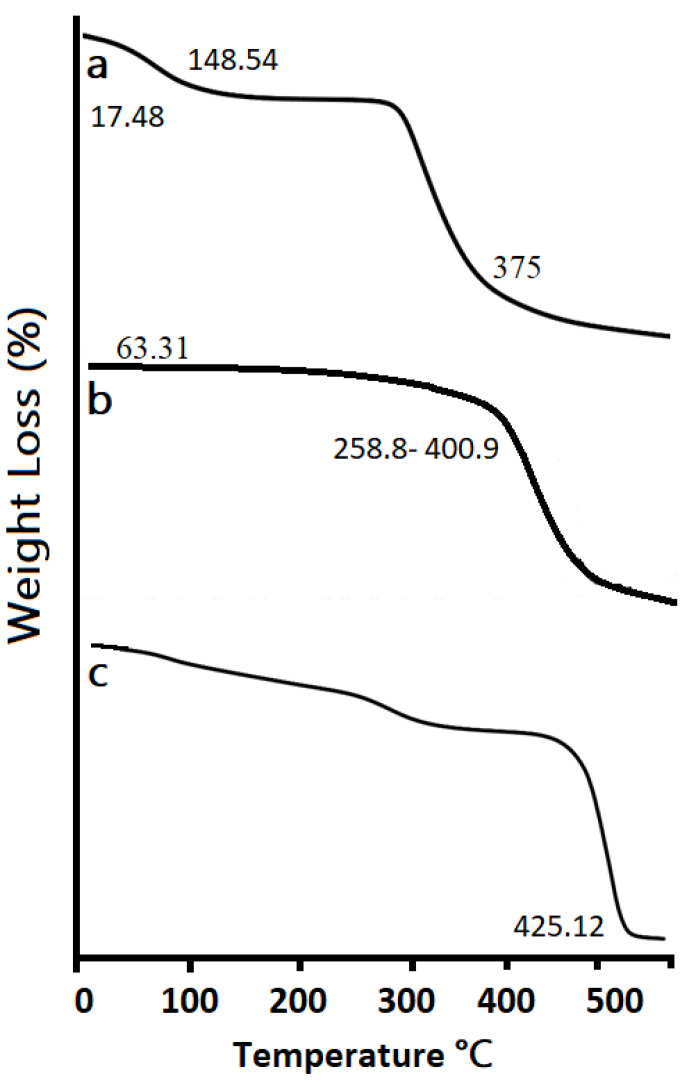
TGA pattern of (**a**) Agarose, (**b**) Pluronic Acid, and (**c**) Optimized formulation.

**Figure 9 pharmaceutics-14-01218-f009:**
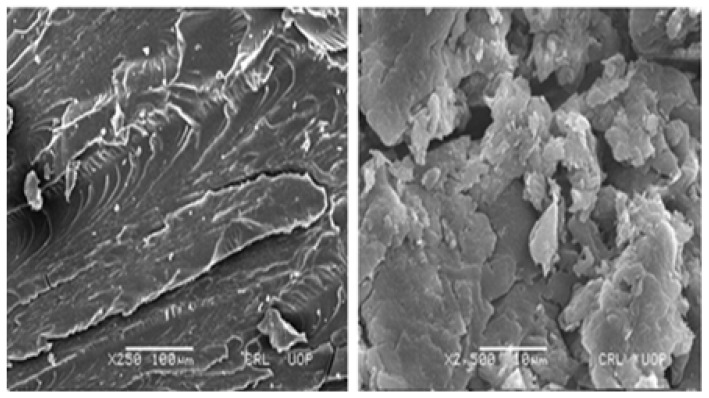
Microphotographs of formulation at different magnifications.

**Figure 10 pharmaceutics-14-01218-f010:**
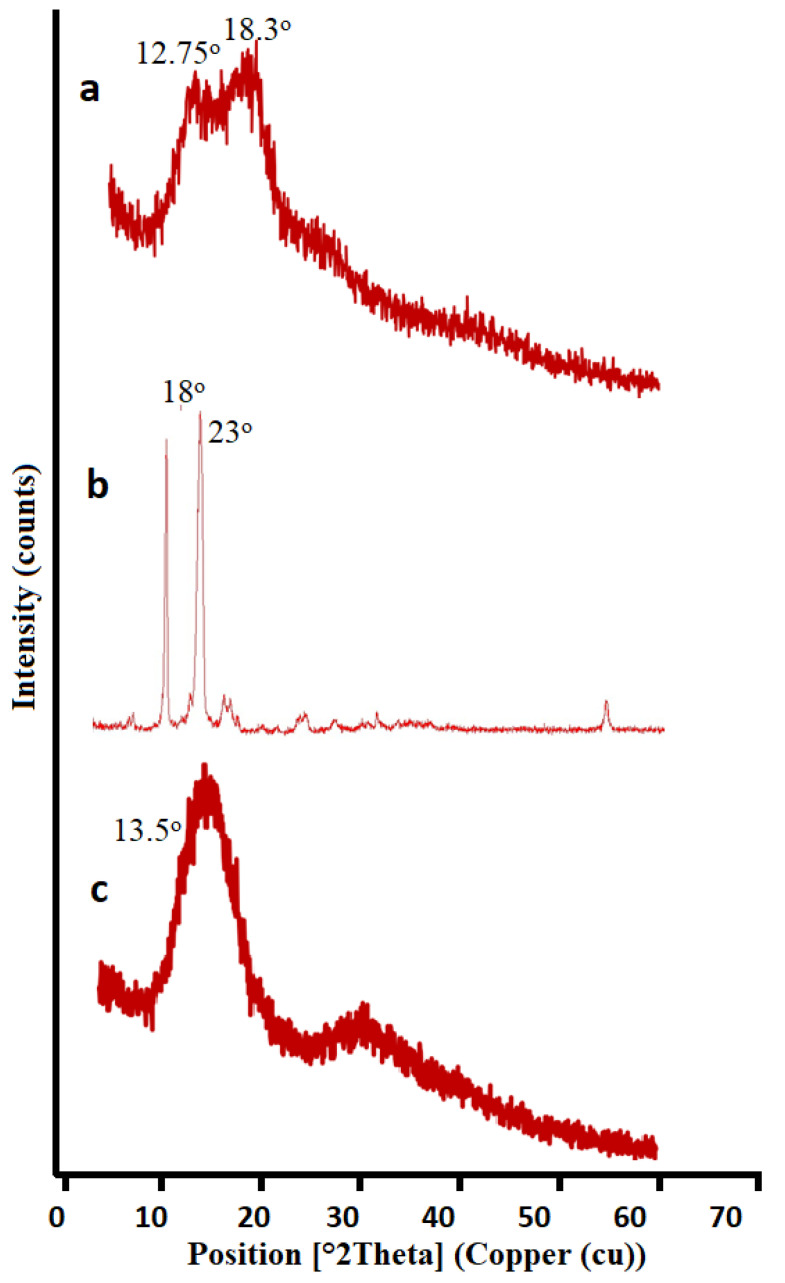
XRD Analysis of (**a**) Agarose, (**b**) Pluronic Acid, (**c**) Optimized formulation.

**Figure 11 pharmaceutics-14-01218-f011:**
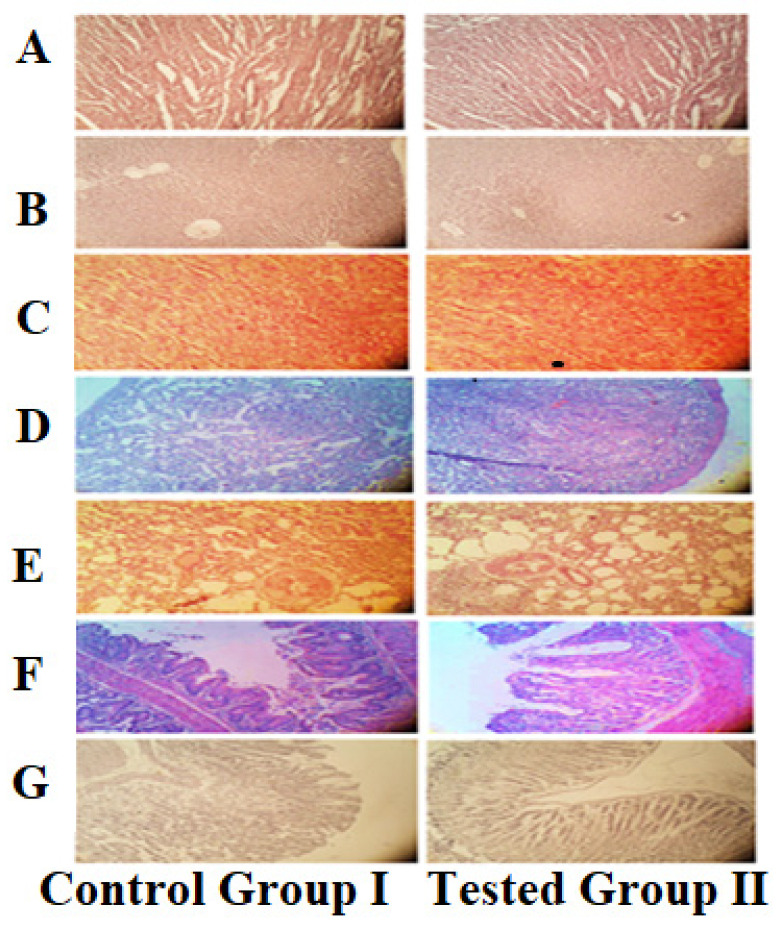
Micrographs of Control and Test group of Rabbit’s Organ Tissues ((**A**) Heart Tissue, (**B**) Liver, (**C**) Spleen, (**D**) Kidney, (**E**) Lungs, (**F**) Intestine, and (**G**) Stomach).

**Table 1 pharmaceutics-14-01218-t001:** Formulation contents of prepared Hydrogels.

Sr. No.	Formulation Code	Agarose (wt%)	Pluronic Acid (wt%)	Methacrylic Acid (wt%)	Benzoyl Peroxide (wt%)	Glutaraldehyde (wt%)
1	AGA 1	0.4	0.8	24	0.4	0.2
2	AGA 2	0.8	0.8	24	0.4	0.2
3	AGA 3	1.2	0.8	24	0.4	0.2
4	AGA 4	0.8	0.4	24	0.4	0.2
5	AGA 5	0.8	1.2	24	0.4	0.2
6	AGA 6	0.8	0.8	20	0.4	0.2
7	AGA 7	0.8	0.8	28	0.4	0.2
8	AGA 8	0.8	0.8	24	0.4	0.4
9	AGA 9	0.8	0.8	24	0.4	0.6

**Table 2 pharmaceutics-14-01218-t002:** Drug loading efficiency (% DLE).

Sr. No.	Formulation Code	Drug Loading Efficiency (% DLE)
1	AGA 1	60.98
2	AGA 2	75.32
3	AGA 3	89.44
4	AGA 4	77.88
5	AGA 5	65.02
6	AGA 6	82.68
7	AGA 7	58.65
8	AGA 8	67.33
9	AGA 9	70.21

**Table 3 pharmaceutics-14-01218-t003:** Results of CP Release from Hydrogel Blend after applying Kinetic Models.

Formulations	Zero-Order	First-Order	Korsmeyer-Model	Higuchi-Model
	(R^2^)	(R^2^)	(R^2^)	(R^2^)
AGA 1	0.9398	0.6777	0.9781	0.9534
AGA 2	0.9078	0.714	0.9787	0.918
AGA 3	0.9339	0.7363	0.9801	0.9355
AGA 4	0.9373	0.6964	0.9764	0.9309
AGA 5	0.927	0.7064	0.9747	0.9236
AGA 6	0.9268	0.7003	0.976	0.9255
AGA 7	0.9461	0.6881	0.9812	0.9589
AGA 8	0.9354	0.7584	0.9459	0.9785
AGA 9	0.9389	0.802	0.9506	0.9813

**Table 4 pharmaceutics-14-01218-t004:** Clinical Findings of Control and Test-Group Rabbits. Results are expressed as Mean ± SD of 3 rats in each group. All values have *p* > 0.05, indicating statistically insignificant difference between two groups.

Findings	Controlled Group	Test Group
	n = 3	n = 3
	Mean ± SD	Mean ± SD
Skin irritation	No	No
Sickness	No	No
Body weight (g) before treatment	1709.33 ± 46.7047	1646.33 ± 40.5504
1st day	1699.67 ± 50.5206	1705.67 ± 40.0042
7th day	1690.67 ± 31.7228	1658.33 ± 41.4769
14th day	1678.33 ± 54.5008	1659.33 ± 39.0171
Water intake (ml) before treatment	245.667 ± 17.0098	225.667 ± 13.0512
1st day	230.667 ± 22.745	248.667 ± 14.7422
7th day	240.333 ± 13.0512	244.667 ± 27.4651
14th day	256.333 ± 22.301	237.667 ± 8.5049
Food intake (g) before treatment	81.3333 ± 4.04145	73.6667 ± 5.1316
1st day	72.6667 ± 4.04145	68.6667 ± 1.52753
7th day	77.6667 ± 5.50757	67.6667 ± 3.78594
14th day	69.3333 ± 4.04145	69.3333 ± 6.50641
Ophthalmic toxicity	No	No
Mortality	No	No

**Table 5 pharmaceutics-14-01218-t005:** Hematology and Biochemical Analysis of Control and Test-Group Rabbits. Results are expressed as Mean ± SD of 3 rats in each group. All values have *p* > 0.05, indicating statistically insignificant difference between two groups.

Hematology	Controlled Group	Test Group
and	n = 3	n = 3
Biochemical Analysis	Mean ± SD	Mean ± SD
Hb (g/dL)	11.76333 ± 0.070238	12.14 ± 0.60506
Total RBCs	5.823333 ± 0.130512	5.98333 ± 0.10017
(3.8–7.9 × 10^6^/mm^3^)		
White cells × 10^3^/cmm	3.603333 ± 0.12741	3.78 ± 0.10536
Lymphocytes (43–80%)	75.59 ± 3.820785	74.7067 ± 1.43151
Monocytes %	3.696667 ± 0.072342	3.82333 ± 0.12503
MCV %	62.824 ± 2.559787	62.93 ± 1.78539
MCHC %	30.54667 ± 1.173641	30.9 ± 1.88364
MCH (pg)	20.03333 ± 0.992791	20.5933 ± 0.78806
Creatinine (mg/dL)	0.946667 ± 0.077675	0.95667 ± 0.10017
AST (10–98 IU/L)	64.37 ± 1.317536	65.1467 ± 1.52821
ALT (55–210 IU/L)	104.1533 ± 0.837158	103.957 ± 1.36749
Total cholesterol	35.68 ± 0.786321	35.5067 ± 0.41259
(10 mg/dL–80 mg/dL)		
Urea (mcg/dL)	51.76 ± 1.17222	52.6133 ± 1.13147
Uric acid	2.566667 ± 0.090738	2.77 ± 0.2816
(1 mg/dL–4.3 mg/dL)		

## Data Availability

Not applicable.

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
