# Peer review of "pH Sensitive Pluronic Acid/Agarose-Hydrogels as Controlled Drug Delivery Carriers: Design, Characterization and Toxicity Evaluation"

_pharmaceutics, 2022, doi:10.3390/pharmaceutics14061218_

Round 1
Reviewer 1 Report
The authors addressed most of my comments. Therefore, the new version of the manuscript is suitable for publication.
Author Response
Dear Reviewer
Thanks for your comments and for accepting our manuscript.
Reviewer 2 Report
Authors have reacted to all queries given
Author Response

(The authors gave the same response as above.)

Reviewer 3 Report
Dear author, in maniuscript pH sensitive pluronic acid/agarose-hydrogels as controlled drug delivery carriers: Design, Characterization and Toxicity Evaluation. Have reported a very interesting research based facts and information it will be very helpful to the researchers, I would like to suggest some pints please revise your manuscript to the following suggested points, I strongly recommend revising this manuscript as follows:
1. Author should redraw the chemical scheme on Chem Draw or other convenient software and add the clear image of hydrogel in the figure number 1 with crosslinking chemical structure.
2. Correct the references some references do not have authors name.
3. Author should add and cite the alginate based recent article in the revised manuscript Advancement of Biomaterial‐Based Postoperative Adhesion Barriers." Macromolecular bioscience 21, no. 3 (2021): 2000395. And Balance of antiperitoneal adhesion, hemostasis, and operability of compressed bilayer ultrapure alginate sponges." Biomaterials Advances (2022): 212825.
4. Author should characterise and include the pluronic and their modification information and cite the appropriate article in the revised manuscript
Author Response
Response Letter
Editor-in-Chief, Date: 4 June, 2022
Pharmaceutics
Manuscript ID: pharmaceutics-1757200
Dear Guest Editor,
Enclosed please find a copy of our revised manuscript entitled “pH sensitive pluronic acid/agarose-hydrogels as controlled drug delivery carriers: Design, Characterization and Toxicity Evaluation” (pharmaceutics-1757200). Our manuscript has been suitably amended based on the constructive comments from reviewer 3. Those comments are all valuable and very helpful for revising and improving our paper, as well as have important guiding significance to our research work. The changes that have been made to the manuscript during the revision process are outlined as enclosed. We hope you would consider our revised manuscript favorably for the publication in Pharmaceutics.
We look forward to your positive reply.
Thank you.
Sincerely yours,
Dr Faisal Raza
School of Pharmacy
Shanghai Jiao Tong University, Shanghai ,200240 P.R. China.
Tel: +86-18521079146
E-mail: faisalraza@sjtu.edu.cn
RESPONSE TO REVIEWERS
REVIEWER 3
Dear author, the manuscript pH sensitive pluronic acid/agarose-hydrogels as controlled drug delivery carriers: Design, Characterization and Toxicity Evaluation.
Have reported a very interesting research-based facts and information it will be very helpful to the researchers, I would like to suggest some pints please revise your manuscript to the following suggested points, I strongly recommend revising this manuscript as follows:
- Author should redraw the chemical scheme on Chem Draw or other convenient software and add the clear image of hydrogel in the figure number 1 with crosslinking chemical structure.
Response: As advised, the chemical scheme was redrawn on Chem Draw indicating cross linked chemical structure. Image of developed hydrogel is indicated in figure number 2.
- Correct the references some references do not have authors name.
Response: References have been corrected as advised.
- Author should add and cite the alginate based recent article in the revised manuscript Advancement of Biomaterial‐Based Postoperative Adhesion Barriers." Macromolecular bioscience 21, no. 3 (2021): 2000395. And Balance of antiperitoneal adhesion, hemostasis, and operability of compressed bilayer ultrapure alginate sponges." Biomaterials Advances(2022): 212825.
Response: Both articles have been cited in revised manuscript as advised.
- Author should characterize and include the pluronic and their modification information and cite the appropriate article in the revised manuscript.
Response: Pluronic acid and developed hydrogel structure has already been characterized using FTIR.
Article has been cited as advised.

This manuscript is a resubmission of an earlier submission. The following is a list of the peer review reports and author responses from that submission.
Round 1
Reviewer 1 Report
The manuscript by Aslam and colleagues deals with the preparation of cross-linked polymeric hydrogels made of agarose, pluronic acid, methacrylic acid and glutaraldehyde. The aim is to obtain a drug delivery system for cytotoxic drugs (e.g. cyclophosphamide). The authors also fully characterized the hydrogel and conducted in vivo toxicity studies in white albino rabbits. The manuscript falls within the aim and scope of Pharmaceutics but not in the one of the special issue to which it was submitted.
I have the following concerns:
- The introduction section has to be rewritten entirely. This is because it lacks any logical flow.
- Also, from the introduction section, the rationale for which these hydrogels have been prepared and their application are not clear. For example, are they intended to be administered orally? It seems so from later parts of the manuscript, but it must be specified.
- The abstract mentions that these hydrogels are prepared to deliver cyclophosphamide to treat non-Hodgkin’s lymphoma. Is administering cytotoxic drugs orally to treat non-Hodgkin’s lymphoma an established clinical practice? Authors should discuss this point.
- The authors prepared nine different formulations by changing the quantities of all excipients, as reported in Table 1. However, the characterization has not been performed on all the formulations prepared. Alternatively, at least the results are not reported for all the formulations. The authors should clarify this aspect.
- The authors reported that the drug was loaded onto hydrogels through the swelling diffusion method. Later, in Equation (2), they mention a theoretical loading. How has it been calculated?
- It is not clear from the materials and methods section how the hydrogels have been administered for the in vivo toxicity studies.
- It is not clear whether the characterization has been performed on all the formulations. Therefore, it is also unclear which formulation the results reported refer to.
- Figure 1A is missing. Figure 1B is of lousy quality and needs to be redone.
- Figure 2 is of lousy quality and needs to be redone. The figure should also indicate (e.g. with arrows) the peaks mentioned in the text.
- Figures 3, 4, 5 and 6 are bad quality and need to be redone.
- Figure 7: the statistical analysis is missing. Any conclusion without statistical analysis has no significance.
- Figure 8: the statistical analysis is missing. Any conclusion without statistical analysis has no significance. Also, it is unclear what formulation the drug release data refer to.
- Figure 9 is of lousy quality and needs to be redone. What formulation has been tested in the toxicity studies?
- The paper lacks any discussion.
- Statistical analysis has been done only for toxicity studies. It has to be also done on data collected from the other experiments.
Minor points:
- Specify the acronyms SIF and SGF in the abstract.
- Line 60-61: medical technology seems not appropriate. Maybe it is better pharmaceutical technology.
- Lines 63-66: it is not clear. Please rewrite.
- In the kinetics equation, pay attention to superscript and subscript.
- Figure 1B caption: the term physical seems inappropriate.
Reviewer 2 Report
This is an exhaustive work work deals with preparation and comprehensive study of the characteristics of cross-linked polymeric blend for controlling delivering of model drug in the subjects suffering form non-hodgkin´s lymphoma.
The material to fabricate blend was agarose, pluronic acid, glutaraldehyde and methcrylic acid.
DSC, FTIR TGA, XRD and SEM approaches were applied for studying characteristics/features of fabricated hydrogels. This values the comprehensiveness of the work.
REMARKS:
NOTE
First paragraph of introduction should include the down below upgrade with following citation.
“Recent advances in high-efficiency drug delivery systems in have been increasing [https://doi.org/10.1038/s41598-021-99678-y]. Here, hydrogels have been introduced and used as suitable materials for medical technology.”
NOTE
Please, try to provide a figure of mechanism of drug delivery in the subject you studied.
NOTE
In conclusion, please provide more of the authors future aims in the scope of enhanced drug delivering system in biological subjects.
Reviewer 3 Report
Dear author, please revise your manuscript to the following suggested points
Prof. Faisal Raza and co-workers have reported very interesting study based on pH sensitive pluronic acid/agarose- hydrogels for controlled drug delivery application I strongly recommend revising this manuscript as follows:
- Authors should include the previous studies regarding agarose-based pH-responsive hydrogels for drug delivery application, author should cite the following article in the revised manuscript: (1) Effect of polyethylene glycol on properties and drug encapsulation–release performance of biodegradable/cytocompatible agarose–polyethylene glycol–polycaprolactone amphiphilic co-network gels. ACS applied materials & interfaces, 8(5), 3182-3192. (2) Degradable/cytocompatible and pH responsive amphiphilic conetwork gels based on agarose-graft copolymers and polycaprolactone." Journal of Materials Chemistry B 3.43 (2015): 8548-8557 (3) Reactive compatibilizer mediated precise synthesis and application of stimuli-responsive polysaccharides-polycaprolactone amphiphilic co-network gels. Polymer, 99, 470-479
- I would like to recommend to the author to change the figure Figure 9: Micrographs of Control and Test group of Rabbit’s Organ Tissues with clear picture of in some pictures seems same picture authors have used in the control and test group please change it and correct it.
- Include the synthesis procedure and characterization of the Pluronic acid/Agarose-co-poly (Methacrylic acid).
- Add the chemical synthetic scheme and hydrogel preparation with the clear digital image of the prepared hydrogel in the revised manuscript.
- In the introduction line number 76-79 authors have discussed about pH-responsive hydrogels, I would like to suggest citing the following pH-responsive hydrogels in the revised manuscript (1) Liquid prepolymer-based in situ formation of degradable poly (ethylene glycol)-linked-poly (caprolactone)-linked-poly (2 dimethylaminoethyl) methacrylate amphiphilic conetwork gels showing polarity driven gelation and bioadhesion." ACS Applied Bio Materials 1, no. 5 (2018): 1606-1619 (2). Self-assembly of partially alkylated dextran-graft-poly [(2-dimethylamino) ethyl methacrylate] copolymer facilitating hydrophobic/hydrophilic drug delivery and improving conetwork hydrogel properties. Biomacromolecules, 19(4), 1142-1153. (3) Dually crosslinked injectable hydrogels of poly (ethylene glycol) and poly [(2-dimethylamino) ethyl methacrylate]-b-poly (N-isopropyl acrylamide) as a wound healing promoter. Journal of Materials Chemistry B, 5(25), 4955-4965.
- Author should rearrange the (Figure 2: FTIR Analysis Pluronic acid, Agarose, MAA, unloaded hydrogel and drug-loaded 306 hydrogel formulation) in a compact and in nice way and indicate the peak assignment, clearly so that readers can easily understand the changes.
- Author should rearrange the (Figure 3: DSC Analysis of (a) Agarose, (b) Pluronic Acid, (c) Optimized unloaded Formulation.) in compact and in nice way and indicate the peak assignment, clearly so that readers can easily understand the changes.
- The author should rearrange the (Figure 4: TGA pattern of (a) Agarose, (b) Pluronic Acid, (c) hydrogel formulation) in compact and in nice way and indicate the peak assignment, clearly so that readers can easily understand the changes.
- I would like to recommend to author to change the figure Figure 9: Micrographs of Control and Test group of Rabbit’s Organ Tissues with a clear picture of in some pictures seems same picture authors have used in the control and test group please change it and correct it.
- Include the synthesis procedure and characterization of the Pluronic acid/Agarose-co-poly (Methacrylic acid).
- Add the chemical synthetic scheme and hydrogel preparation with the clear digital image of the prepared hydrogel in the revised manuscript.